# Discovery of a New Microbial Origin Cold-Active Neopullulanase Capable for Effective Conversion of Pullulan to Panose

**DOI:** 10.3390/ijms23136928

**Published:** 2022-06-22

**Authors:** Meixing Wang, Huizhen Hu, Buyu Zhang, Yang Zheng, Pan Wu, Zhenghui Lu, Guimin Zhang

**Affiliations:** 1College of Life Science and Technology, Beijing University of Chemical Technology, Beijing 100029, China; wangmeixing@buct.edu.cn; 2State Key Laboratory of Biocatalysis and Enzyme Engineering, School of Life Sciences, Hubei University, Wuhan 430062, China; jenny_8729@163.com (H.H.); zhangby@163.com (B.Z.); yzheng0115@163.com (Y.Z.); wupan@hubu.edu.cn (P.W.); zhenghuilu@hubu.edu.cn (Z.L.); 3Yunnan Province Engineering Research Center for Functional Flower Resources and Industrialization, College of Landscape Architecture and Horticulture Sciences, Southwest Forestry University, Kunming 650224, China

**Keywords:** neopullulanase (NPase), pullulan, panose production, high-efficiency

## Abstract

Panose is a type of functional sugar with diverse bioactivities. The enzymatic conversion bioprocess to produce high purity panose with high efficiency has become increasingly important. Here, a new neopullulanase (NPase), Amy117 from *B. pseudofirmus* 703, was identified and characterized. Amy117 presented the optimal activity at pH 7.0 and 30 °C, its activity is over 40% at 10 °C and over 80% at 20 °C, which is cold-active. The enzyme cleaved α-1, 4-glycosidic linkages of pullulan to generate panose as the only hydrolysis product, and degraded cyclodextrins (CDs) and starch to glucose and maltose, with an apparent preference for CDs. Furthermore, Amy117 can produce 72.7 mg/mL panose with a conversion yield of 91% (*w*/*w*) based on 80 mg/mL pullulan. The sequence and structure analysis showed that the low proportion of Arg, high proportion of Asn and Gln, and high α-helix levels in Amy117 may contribute to its cold-active properties. Root mean square deviation (RMSD) analysis also showed that Amy117 is more flexible than two mesophilic homologues. Hence, we discovered a new high-efficiency panose-producing NPase, which so far achieves the highest panose production and would be an ideal candidate in the food industry.

## 1. Introduction

Functional oligosaccharides are widely used in the food industry [1]. Panose is a trisaccharide which is composed of a maltose molecule bonded to a glucose molecule through an α-1, 6-glycosidic bond and has the potential to be used as a functional oligosaccharide in the food industry. Panose is widely used as an anti-caries sweetener, anti-fading agent and anti-oxidant [2]. As this carbohydrate cannot be fermented by oral microorganisms, it has substantial low or caries-resistant properties, in addition to acting as a sweetener [3]. Panose is also considered as a prebiotic, which can stimulate the growth of beneficial microorganisms (such as *bifidobacteria* and *lactobacillus*) and inhibit the growth of unnecessary microorganisms (such as *Salmonella*) [4]. The health functions of panose will be paid more and more attention.

There are two main kinds of pullulan hydrolases that can produce panose by the enzymatic hydrolysis of pullulan, pullulan hydrolase type I (EC 3.2.1.135) and pullulan hydrolase type III with a high sequence identity [5]. Pullulan hydrolase type III hydrolyzes pullulan to obtain a mixture of panose, maltose and maltotriose [5]. Obviously, this hydrolysate needs a complex sugar purification process to obtain pure panose. In contrast, the pullulan hydrolase type I (EC 3.2.1.135), also called neopullulanase (NPase), can cut pullulan to generate panose as the only product [6] and, thus, has the potential application of producing high purity panose. The catalytic reaction diagram is shown in Appendix A. In order to meet the requirements of the application, it is necessary to discover new NPases which can produce panose efficiently and demand further study to explore their mechanisms and applications.

NPases have been classified as part of the glycosyl hydrolase (GH) family 13 [7,8], also known as the α-amylase family, which is the largest sequence-based glycoside hydrolase family and combines many different enzyme activities and substrate specificities to act on α-glycosidic bonds [9]. These enzymes are called poly-specific enzymes and hydrolyze at least one of the following three substrates, including pullulan, soluble starch or cyclodextrins (CDs). Among the three substrates, the enzymes hydrolyzed CDs and soluble starch into glucose, maltose, and/or higher malt-oligosaccharides [6,10,11], and hydrolyzed pullulan into panose. Typical α-amylase structures contain a central (α/β)8-barrel domain and a C-terminal domain, while those poly-specific enzymes also contain a distinct N-terminal domain. Two molecules of the protein form a domain-swapped dimer in which the N-terminal domain of one molecule is involved in extensive interactions with the (α/β)8-barrel domain of the adjacent molecule. Earlier studies showed that the NPase from *Bacillus stearothermophilus* (bsNpl) generated panose from the hydrolysis of pullulan, hydrolyzed CDs and soluble starch to glucose, maltose, and higher malto-oligosaccharides [12]. However, the NPase of *Geobacillus stearothermophilus* was an exception, in addition to panose, maltose, and glucose were also produced from the pullulan hydrolysis and showed a higher substrate specificity for pullulan than the CDs and soluble starch [12]. The NPase of *Laceyella sacchari* (Amy92) hydrolyzes pullulan, CDs, and soluble starch with comparable activity, the product of pullulan hydrolysis was panose, and the major products of CD hydrolysis were glucose and maltose [10,11]. Afterward, the researchers found that Amy132 (environmental DNA), Amy98 (*Anoxybacillus flavithermus*), and Amy29 (environmental DNA) all harbored NPase activities [13,14]. Amy132 and Amy98 had a high sequence identity and, thus, exhibited similar substrate specificity. Both can hydrolyze pullulan to form panose, while they can degrade soluble starch and CDs into glucose, maltose and maltotriose (occasionally with higher malto-oligosaccharides), with an obvious hydrolysis preference for CDs. The preferred substrate for Amy29 was pullulan, which was degraded into panose, and it was the only enzyme that had no detectable activity on the CDs. Recently, three enzymes, including the NPase of *L. mucosae* LM1 [15], the maltogenic amylase CoMA from *Corallococcus* sp. EGB [16] and the α-amylase Rbamy5 from *Ruminococcus bromii* ATCC 27255 [17], were reported to hydrolyze pullulan exclusively to panose.

Although a few of the NPases with pullulan-hydrolytic properties and panose as the only product have been described, the conversion ratios of panose have seldom been investigated. Only Rbamy5 was reported to transform 5% pullulan (*w/v*) into panose with an 80% conversion rate [17], so few enzymes contain the conversion rate data, limiting their large-scale commercial application. A suitable NPase is the key factor for the efficient production of high purity panose in the bioprocess of enzymatic conversion, and it demands further research to study its properties and applications.

In this study, we first obtained a suspected α-glycosidase gene, *Amy117*, from *B. pseudofirmus* 703. Through the heterologous expression in *E. coli*, the function of the gene-expressed protein was explored. Its physicochemical properties, including temperature and pH preference, substrate specificity, and its pullulan-hydrolytic efficiency to panose were investigated, aiming to provide a firm basis for the further application of this enzyme in the food industry.

## 2. Results

### 2.1. Sequence Analysis, Expression, and Purification of Amy117

The gene (*Amy117*) from *B. pseudofirmus* 703 contains the full-length coding region, including a 1770 bp open reading frame ((ORF); Appendix A) without the signal peptide-encoding sequence (predicted by SignalP 4.1). The Amy117 contains 590 amino acid residues, with a theoretical pI of 5.1 and a predicted molecular weight of 68.7 kDa, respectively. The *Amy117* gene sequence was submitted to the NCBI database, for which the accession number was MK882930, and its protein sequence showed a 96% identity with the α-glycosidase from *B. pseudofirmus* OF4 (WP_012959815.1; it was sequenced without functional research).

The sequence alignment showed that the Amy117 contained three strictly conserved catalytic residues (Asp333, Glu362 and Asp429), and carried four sequences conserved among the enzymes belonging to the GH13 family [18], including: I: 241-246 DAVFNH; II: 329-337 GWRLDVANE; III: 362-365 EIWH; and IV: 424-429 LLGSHD, respectively, as shown in Figure 1. It presented a high sequence identity with some of the characterized members in the GH13 family (Figure 1; Appendix A), which are Maltogenic amylase (MAase) from *Parageobacillus caldoxylosiyticus* (ACN79585.1, 64%) [19], followed by MAase from *G. thermoleovorans* (AFM43699.1, 63%) [20], BSMA from *G. stearothermophilus* (AAC46346.1, 63%) [21], Cyclomaltodextrinase (CDase) Amy98 from *A. flavithermus* (AAX29991.1, 63%) [11], MAase ThMA from the *Thermus* strain (AAC15072.1, 62%) [22], NPase from *G. stearothermophilus* (AAK15003.1, 61%) [12], and CDase Amy132 from environmental DNA (ACA48225.1, 61%) [14].

The amylase gene (*Amy117*) was cloned from *B. pseudofirmus* 703 and expressed in *E. coli* BL21 (DE3), then the recombinant Amy117 (pET28a-*Amy117*) was induced for expression, purification, and desalination. The molecular weight of the purified Amy117 was estimated to be 68.0 kDa by SDS-PAGE (Appendix A), corresponding to the calculated value of the theory.

### 2.2. Effects of Temperature and pH on the Activity and Stability of Amy117

The optimum temperature of Amy117 towards α-CD was 30 °C, and it can exhibit over 80% relative activity at 20 °C (Figure 2A). Even at 10 °C, the relative activity of Amy117 remained at 40%. This means that Amy117 is a cold-active enzyme, which allows both energy and cost savings. As exhibited in Figure 2B, Amy117 remained stable after 30 min pre-incubation at 20 °C and 30 °C, and retained 60% residual activity after 30 min pre-incubation at 40 °C, but displayed rapid inactivation above 50 °C.

The purified Amy117 has the strongest activity at pH 7.0 in 50 mM sodium phosphate buffer; it can present over 50% activity from pH 5.5–8.0 (Figure 2C). As shown in Figure 2D, Amy117 was stable in the range of pH 7.0–10.0 and maintained more than 80% of its maximum activity after incubation at 4 °C for 16 h, but lost most of its activity at pH 12.0. Therefore, Amy117 is a neutral enzyme with excellent pH stability.

### 2.3. Substrate Specificity of Amy117

Extensive studies showed that the α-amylases can act on α-glycosidic bonds [9], thus, the substrate specificity of Amy117 was investigated on eight glycoside compounds, including α-CD, β-CD, γ-CD, pullulan, soluble starch, amylopectin from potato, amylopectin from maize, and amylose from potato. Amy117 showed high activity towards the CDs, followed by pullulan and soluble starch, but it had no activity towards the amylopectins and amylose (Table 1). The results further proved that Amy117 has the activity of amylase, however, with an apparent preference for CDs’ hydrolysis. The kinetic parameters of Amy117 for the natural products, pullulan and soluble starch, were measured. The *K_m_*, *V_max_*, and *k_cat_* of Amy117 against pullulan are 4.62 μM, 52.01 μM/s, and 33.77 s^−1^, respectively. The *K_m_*, *V_max_* and *k_cat_* of Amy117 against starch are 13.85 μM, 53.07 μM/s, and 34.46 s^−1^, respectively.

### 2.4. Action Modes of Amy117 on Various Substrates

To further understand the hydrolytic properties of Amy117, we then investigated the hydrolytic products of Amy117 on CDs in detail by HPLC–ELSD (Figure 3; Appendix A). The results showed that Amy117 degraded CDs to maltose and glucose (Figure 3), and the ratio of maltose to glucose for α-CD, β-CD and γ-CD were 2.584, 2.680 and 2.954, respectively (Appendix A). This result suggests that Amy117 could degrade CDs to produce a higher percentage of maltose.

CDs are known to be synthetic, non-natural substrates; we further studied the hydrolytic products of the natural substrate soluble starch. Amy117 presents relatively low activity for soluble starch hydrolysis (Table 1), hence, amylase BLA was used for comparison. The hydrolytic products from 1% (*w/v*) soluble starch by Amy117 and BLA were analyzed using TLC, respectively (Figure 4A). The results showed that BLA could degrade soluble starch to maltose, maltotriose, and a number of oligosaccharides (Figure 4A, second column), whereas Amy117 could hydrolyze soluble starch to glucose and maltose (Figure 4A, third column). It is interesting to note that after the BLA hydrolysis of soluble starch, Amy117 could further completely degrade the maltotriose and oligosaccharides to more glucose and maltose (Figure 4A, fourth column). Taken together, Amy117 can assist the common amylase BLA in the hydrolysis of maltotriose and oligosaccharides into maltose and glucose completely during the soluble starch hydrolysis process, though Amy117 presents relatively low activity for soluble starch hydrolysis.

Pullulan polysaccharide is a kind of extracellular water-soluble mucopolysaccharide that is fermented from corn. The polysaccharide is mainly composed of maltotriose linked by α-1, -6 glycoside bonds, and it is known as a pollution-free plastic as it can be degraded and utilized by microorganisms in nature. Pullulan polysaccharide is one of the four new food additives. It can be used as a film-coating agent and a thickener in candy, chocolate coating, films, compound condiments, and fruit and vegetable juice beverages [23]. We therefore investigated the pullulan hydrolysate by Amy117. The results showed that an individual BLA has no activity on pullulan (Figure 4B, second column), while an individual Amy117 could hydrolyze pullulan to form a single product different to glucose, maltose, and maltotriose (Figure 4B, third column). Moreover, the same product was detected with the BLA and Amy117 together (Figure 4B, fourth column). Then, we performed Liquid chromatography–Mass spectrometry (LC–MS) to determine the molecular weight of the hydrolytic product from pullulan, which was 527.16 (Appendix A). Given that the molecular weight of maltotriose, isomaltotriose, and panose is 527.16, we then used TLC and HPLC to detect the hydrolytic products in detail. The TLC results revealed that the hydrolytic product of Amy117 from pullulan was more likely to be panose (Figure 4C), which was confirmed by HPLC–ELSD, that the product was panose (Figure 4D). Combined with the amino acid sequence identity and the hydrolytic properties of Amy117, we judged that it belongs to NPase, since all of the NPases hydrolyze pullulan to form panose as the main final product.

### 2.5. Production of Panose by Amy117

Although it has been reported that the NPases hydrolyze pullulan to form panose, the hydrolytic efficiency of these NPases have seldom been investigated. In order to explore the efficiency of Amy117 for panose production, we did a substrate concentration gradient experiment, with pullulan as the substrate (Table 2). The final concentration of the enzyme used for the reactions at the highest substrate concentration was 0.32 mg/mL (19.9 U/mL towards pullulan). The results showed that the absolute weight of panose increased with the increase in the substrate concentration. To be specific, the concentrations of panose were 10.42 mg/mL, 40.33 mg/mL, 72.74 mg/mL, and 72.72 mg/mL, corresponding to 10 mg/mL, 40 mg/mL, 80 mg/mL, and 100 mg/mL of pullulan, respectively. That is, the conversion yields (*w*/*w*) were 104.2%, 100.8%, 90.9%, and 72.7%, respectively. Therefore, Amy117 could produce about 73 mg/mL panose with a conversion yield of 91% (*w*/*w*), based on 80 mg/mL pullulan, which would be the ideal system. It has a greater advantage than Rbamy5, with a conversion yield of 80% based on 5% pullulan, the only panose yield that was reported before [17].

### 2.6. Structure Modeling of Amy117

The simulated structure analysis showed that Amy117 is a dimer (Figure 5), and each monomer possesses the typical domain structure of GH13_20, including the N-terminal carbohydrate-binding module CBM34 and the catalytic module [24]. Domain N consists of 11 antiparallel β-chains and is specific for neopullulanase and other homologues [25]. The catalytic module is composed of an A-, B- and a C-terminal domain with catalytic sites Asp333-Glu362-Asp429. Two monomers contact with each other at the surfaces of domains A, B, and N to form the dimer, in the same way as left and right hands interlace and clasp (Figure 5A). Domain A and part of B of Mol-1 and domain N of Mol-2 form a narrow active cleft, which is slightly curved and open at both ends (Figure 5B,C). The substrate binds to the cleft of the catalytic pocket and is catalyzed by the catalytic residues. Domain N contributes significantly to the definition of the cleft structure. That is to say, domain N of Mol-2 blocks part of the active pocket of Mol-1, which may explain why Amy117 presented high activity towards the small molecule substrates CDs, but negligible activity for natural polysaccharide soluble starch. Thus, the dimer formation may be thought to be essential for these homologous enzymes to exhibit the unique substrate specificity.

We used analytical ultracentrifugation (Beckman Coulter; Brea, CA, USA) to determine the oligomeric state of Amy117 in a 50 mM Na_2_HPO_4_-NaH_2_PO_4_ buffer (pH 7.0), and ThMA as a control. Data analysis showed that 24.3% of Amy117 existed in the monomeric form, 68.7% of Amy117 existed in the dimerized form, and only 3.3% existed as a tetramer (Figure 5D). Meanwhile, 77.3% of the ThMA existed in the dimerized form and 10.5% existed as a tetramer (Figure 5D) in a 50 mM Na_2_HPO_4_-NaH_2_PO_4_ buffer (pH 7.0).

### 2.7. Molecular Dynamic (MD) of Amy117 and Its Homologues

The Amy117, and two characterized homologues with more than 60% sequence identity, ThMA and BsNp (PDB: 1GVI and PDB: 1J0H) [22,26], were investigated for structural conformational fluctuation. The MD simulation method calculates the behavior of molecules over time, and it can differentiate between both the quality and stability of similar models. According to Pikkemaat et al., the stability of protein can be analyzed using MD study by evaluating the RMSD of the protein structure [27,28]. The RMSD represents the variation of the α-carbon atom in the structure over time; the larger the RMSD value or fluctuation is, the more flexible the protein structure is.

Based on the simulation results presented in Figure 6, each structure showed some increment in the RMSD values during the simulation. The floating change in the RMSD showed the structural changes during the simulation. The RMSD value of Amy117 increased from 1.5 Å to 2.5 Å in the first 14 ns, then kept stable at about 2.5 Å until 24 ns, and then increased to 3–3.5 Å until the end of the simulation. By contrast, the RMSD values of the two homologues fluctuated to 2.5 Å in the first 14 ns, then kept stable with an average of 2.5 Å until the end of the simulation. Our results showed that Amy117 endured more conformational changes during the MD simulations, as shown in Figure 6. The increasing value of the RMSD in the structure is due to enhanced motions between the atoms. Amy117 showed stronger conformational flexibility than the two homologues.

## 3. Discussion

In this article, an NPase Amy117 was cloned and functionally expressed in *E. coli*. The highest activity of Amy117 was at pH 7.0 and 30 °C. It has the property of being cold-active, its activity is over 40% at 10 °C and over 80% at 20 °C, and the degradation product of pullulan is the only panose with an excellent conversion rate. Thus, it has great potential in industrial applications.

The sequence alignment of the protein primary structure showed that Amy117 shares more than 60% identity with some other of the GH13 family enzymes (Appendix A), see 2.1 for details. It is worth noting that such high identity has different names, MAase and CDase, which is confusing. In particular, the first two enzymes in Appendix A share the highest identity with Amy117; since they have not explored the degradation of pullulan, their names as maltogenic amylases are even more suspicious. Furthermore, we did multiple sequence alignments of Amy117 and all of the other reported enzymes that produced panose as the only product from pullulan (Appendix A). Our results showed that the types of enzymes that can hydrolyze pullulan to produce panose as the only product are diverse, and the identity with Amy117 ranges from 63% to 26%. The reports show that not all of the enzymes that hydrolyze pullulan to produce the only panose product are named NPase, and not all enzymes named NPase can only produce panose from the hydrolysis of pullulan.

Summarizing the results of multiple sequence alignment (Appendix A) in this article, we find that the CDases (EC 3.2.1.54) and MAases (EC 3.2.1.133) share more than 60% homologies with the NPase (EC 3.2.1.135), Amy117. The confusing thing is that the three enzymes (CDases, MAases, and NPases) have different enzyme codes and names with such a high identity. In fact, Lee et al. have already stated that the reason why the different researchers independently named the enzymes that they found with any of these three names is mainly based on the substrates that have been selected [8]. For example, it was named CDase since the researchers observed a preference for CDs over other starch materials (such as amylose and pullulan), it was named NPase because of the hydrolysis of pullulan to generate panose, and it was named MAase because of the production of maltose as the main product from CDs and starch. Therefore, they proposed that these three poly-specific enzymes should be classified with the same name and enzyme code to avoid confusion [8], and this was supported by our studies.

Amy117 prefers to show as cold-adapted. Whether the product of degrading pullulan is the only panose or not, the optimum temperature of almost all of these compared enzymes is middle or high temperature, that is, 50–80 °C, with amino acids sequence identity with Amy117 ranging from 64% to 26%. Apart from the enzyme from *Lactobacillus mucosae* LM1 [15], the optimum temperature is 37 °C, which is also higher than 30 °C (Appendix A). The guarantee of high temperatures in industrial production requires a large amount of energy supply. Cold-adapted enzymes can catalyze reactions at room, or even lower temperatures, without heat input into the system; therefore, the ideal biochemical results can be achieved by using a small amout of cold-adapted enzymes, such as Amy117. This can be the basis for developing more cost-effective process qualifications.

The narrow active cleft of Amy117 determines its high activity towards the small molecule substrates CDs. We proposed that the smaller the substrate, the easier it was to enter the narrow active cleft of Amy117, resulting in the highest activity for the smallest substrate, α-CD. On the contrary, the activity of Rbamy5 towards CDs is γ-CD > β-CD > α-CD, because it lacks the N-terminal domain, resulting in a wide and shallow active pocket. This may mean that the smaller substrates are more difficult to stably bind to the active pocket of Rbamy5. Although the characteristics of some NPases have been reported, the pullulan-hydrolytic efficiency of these enzymes has rarely been investigated, which limits their large-scale application in the production of panose, except that Rbamy5 was verified to have a conversion yield of 80%, based on 5% pullulan [17]. Here, Amy117 hydrolyzed pullulan to form panose as the only product, similar to most of the reported NPases, and so on [13,15,17,29,30]. We investigated the conversion ratios of panose by Amy117 and found it was a new high-efficiency panose-producing NPase. Specifically, a substrate concentration gradient experiment showed that Amy117 has high pullulan-hydrolytic efficiency, which produced 72.7 mg/mL panose with a conversion yield of 91% based on 8% pullulan, which is so far the highest report of the production of high purity panose from pullulan by a NPase. Considering the high market price of panose, Amy117, which exhibits high catalytic activity and efficient productivity, should bring benefits to the food industry. In addition, the addition of 1M KCl or NaCl did not affect the activity of Amy117 with pullulan as the substrate, which means that Amy117 can tolerate high concentrations of salt ions. Therefore, Amy117 will have good market application potential, because of its excellent catalytic properties.

We tried to analyze the mechanism of cold-adapted characteristics of Amy117 using structure analysis. The simulated structure of Amy117 is a dimer, in the same way as left and right hands interlace, with four classical domains in a monomer. We speculated that the low-temperature properties of Amy117 may be related to its flexibility at low temperatures, and compared the content of these amino acids in Amy117 and the other two proteins, ThMA and BsNp (PDB: 1GVI and PDB: 1J0H) [22,26], the optimum temperature is 60 °C and 55 °C, respectively. The comparison results are shown in Table 3. According to the data, Amy117 contains less Arg and lower Arg/(Arg + Lys), more Val, Ile, and Thr, more Asn and Gln. Arginine has a larger side chain than the amino acids with the same charge [31], and the hydrophobic effect and ionic interaction provided by the side chain help to improve the thermal stability of the protein. Therefore, Arg and Arg/(Arg + Lys) appear more frequently in thermophilic enzymes [32]. In the α-helix, the amino acids with β-branched chains (Val, Ile, Thr) produce greater conformational tension and are not conducive to the stability of the helix, so they appear less frequently in thermophilic enzymes [33,34,35]. Furthermore, Asn and Gln are heat-sensitive amino acids, and they are easily deaminated at high temperatures, so their frequency in thermophilic proteins is low [36,37]. So, Amy117, with characteristic amino acid frequency, exhibits flexibility at low temperatures.

## 4. Materials and Methods

### 4.1. Plasmids, Strains, Chemicals, and Media

The *B. pseudofirmus* 703 strain was screened in our previous study [38] and deposited at the China Center for Type Culture Collection (CCTCC) with accession number: CCTCC AB 2019311. The *Escherichia coli* DH5α and BL21 (DE3), and the expression vector pET28a were obtained from Stratagene (La Jolla, CA, USA).

The synthesis of the primers and the DNA sequencing were performed in Sangon Biotech Co. Ltd. (Shanghai, China). The Ex Taq DNA polymerase, restriction enzymes, and T4 ligase were purchased from Takara (Dalian, China). The analytical α-cyclodextrin (α-CD), β-cyclodextrin (β-CD), γ-cyclodextrin (γ-CD), pullulan, soluble starch, amylopectin from potato, amylopectin from maize, amylose from potato, panose, isomaltotriose, glucose, maltose, maltotriose, and dextrin (purity > 99%) were obtained from either Sigma–Aldrich (St. Louis, MO, USA) or Aladdin (Shanghai, China). The other analytical chemicals and reagents were purchased on the market. The restriction enzymes and DNA ligase were from New England Biolabs (NEB, Ipswich, MA, USA).

### 4.2. Cloning and Sequence Analysis of an α-Glycosidase or Putative NPase Gene

The *B. pseudofirmus* 703 genomic DNA was extracted and used to amplify Amy117 after being cultured at 37 °C overnight in Horikoshi-I medium [38]. A pair of primers, Amy117-F (5’ CGCGGATCCCAAAAGGAAGCCATTTATCACCGTC 3’) and Amy117-R (5’ CCGCTCGAGTTAAAGCTTCTTTAAAACAATGGCTTGTT 3’) for Amy117 amplification, were designed, based on the genome sequence of *B. pseudofirmus OF4* (WP_012959815.1), the strain with the closest sequence to *B. pseudofirmus 703*. The restriction enzymes were *Bam*HI and *Xho*I.

The ExPASy Proteomics tools (http://www.expasy.org/tools/, accessed on 1 May 2021) were used to analyze the nucleotide and deduce the amino acid sequences. The identity of the protein sequence was analyzed by BLAST. Clustal X performed multiple sequence alignment through default parameters, and the alignment result was exported through ESPript. The SignalP 4.1 server (http://www.cbs.dtu.dk/ services/SignalP, accessed on 1 May 2021) was used to identify signal peptide.

### 4.3. Expression and Purification of the Recombinant α-Glycosidase or Putative NPase in E. coli

The α-glycosidase gene (*Amy117*) was ligated into vector pET28a. The recombinant vector (pET28a-*Amy117*) was transformed into *E. coli* BL21 (DE3) for the expression of the protein, according to the standard expression procedure of *E. coli* [39], that is, the concentration of IPTG was 0.5 mM and the induction time was 18 h at 18 °C. 

All of the protein purification steps were preferably carried out at 4 °C. The culture was collected by centrifuging and the cell was resuspended in 50 mM Tris-HCl (pH 7.5) buffer for lysis by ultrasonic wave (Shanghai Jingxin Industrial Development Co. Ltd. Shanghai, China). The conditions of the sonication to lyse the cells in this paper were sonication for 2 s, pause for 4 s, and cycle for 10 min. The cell lysis supernatant was collected and loaded on a nickel affinity column (His60 Ni Superflow resin from Takara Biomedical Technology Co. Ltd. Dalian, China), then the eluate was desalted on a GE HiTrap Desalting column (GE Heathware, Marlborough, MA, USA), equilibrated with 50 mM sodium phosphate (Na_2_HPO_4_-NaH_2_PO_4_) buffer (pH 7.0). The homogeneity and molecular weight of the protein were evaluated by 12% SDS-PAGE.

### 4.4. Temperature and pH Preference, Substrate Specificity, and Catalytic Constants of the α-Glycosidase or Putative NPase Amy117

#### 4.4.1. Temperature Preference of Amy117

The determinations of optimum temperature and pH were described in our previous report [39] with some modifications. The impacts of temperature on the Amy117 activity were determined within the scope of 4 °C to 80 °C in 50 mM sodium phosphate buffer (pH 7.0), α-CD was used as the substrate. For thermal stability, the purified Amy117 was incubated in 50 mM sodium phosphate buffer (pH 7.0) at different temperatures (20 °C, 30 °C, 40 °C, 50 °C, 60 °C) for 30 min, and then the residual activities were determined by the standard method.

#### 4.4.2. Temperature Preference of Amy117

The optimum pH of Amy117 was detected in different buffers (50 mM) within the range of pH 5.0–13.0. The buffers applied were sodium phosphate buffer (5.0–8.0), Tris-HCl (8.0–10.5), Na_2_HPO_4_-NaOH buffer (10.5–12.0), and KCl-NaOH buffer (12.0–13.0), and α-CD was used as the substrate. In order to determine the stability of the pH, the enzymes were incubated at different pH values, ranging from 5.0 to 13.0 at 4 °C for 16 h, and then the residual activities were analyzed by the standard method.

#### 4.4.3. Substrate Specificity of Amy117

The substrate specificity of Amy117 was detected at 30 °C in 50 mM sodium phosphate buffer (pH 7.0), using different substrates for 30 min. The substrates (1%, *w/v*) included α-CD, β-CD, γ-CD, pullulan, soluble starch, amylopectin from potato, amylopectin from maize, and amylose from potato (Sigma–Aldrich, St. Louis, MO, USA) or Aladdin, Shanghai, China) and the amount of the reducing sugars released was determined by the 3, 5-dinitrosalicylic acid (DNS) method [40]. The enzyme activities of Amy117 were evaluated by the DNS method, in which glucose was used as the reference. The specific activity of the protein against different substrates is different, resulting in different amounts of enzyme added. The reaction under each substrate will be pre-experimented to determine the amount of enzyme added. Briefly, the mixture containing a 50 mM Na_2_HPO_4_-NaH_2_PO_4_ (pH 7.0) buffer, 1% (*w/v*) natural substrate, was pre-warmed at 30 °C for 5 min. Subsequently, the enzyme solution was added to the pre-warmed solution and incubated at 30 °C for 30 min. The reaction was terminated by the addition of the DNS. After boiling for 5 min in a water bath, the absorbance was measured at 540 nm. Using glucose as the standard, the amount of enzyme that released 1 μM of reducing sugar per minute under the measured conditions was called one unit of α-glycosidase. Taking bovine serum albumin (BSA) as the standard, the concentration of the protein was determined by the Bradford method [41].

#### 4.4.4. Catalytic Constants of Amy117

The catalytic constants of Amy117 such as *K_m_*, *V_max_* and *k_cat_* were determined by using 1.54 μM enzyme and varying concentrations of pullulan (0.5–5.0 mg/mL) and soluble starch (1.0–10.0 mg/mL). All the measurements were done in triplicate. Finally the *K_m_*, *V_max_* and *k_cat_* were established by nonlinear regression using the GraphPad Prism software. 

### 4.5. Analysis of Composition the Enzymatic Hydrolysis Products by HPLC and TLC 

The α-CD, β-CD, γ-CD, and pullulan were hydrolyzed by the purified Amy117, then the qualitative analysis was carried out by high-performance liquid chromatography installed with evaporative light scattering detection (HPLC–ELSD) for the hydrolytic products. The glucose, maltose, maltotriose, panose, and isomaltotriose were used as the standards. The 1% (*w/v*) substrate was hydrolyzed at 30 °C for 4 h. Then, the quantitative analysis was carried out with Agilent 1260 equipped with G1311X quaternary gradient pump, G4218A ELSD. The separation was performed on Agilent Zorbax Carbohydrate Analysis Column (150 mm × 4.60 mm, 5 microns) (Agilent, Palo Alto, CA, USA). The mobile phase used was acetonitrile/water (75:25, *v/v*) with a flow rate of 1 mL/min. All of the samples were filtered through a 0.22 μM Millipore membrane before use.

The purified enzymes, Amy117 or *B. licheniformis* α-amylase (BLA) [42], were incubated with 1% (*w/v*) of different substrates (pullulan or soluble starch) at 30 °C in 50 mM sodium phosphate buffer (pH 7.0), then boiled for 10 min to terminate the enzyme activity, and the hydrolytic products were then determined using Thin Layer Chromatography (TLC) [43]. In short, the hydrolytic products were placed on a Silica gel plate (Merck, Darmstadt, Germany) and then infiltrated and expanded for 1–2 h in the solvent system containing n-butanol-acetic acid-water (3:2:1, *v/v*), then, dried for 3–5 min at 50 °C. Then, the plates were sprayed with the solution (sulfuric acid/absolute ethanol, 1:19, *v/v*) and dried for 15 min at 90 °C. The glucose, maltose, maltotriose, panose, and isomaltotriose were used as the standards.

### 4.6. Panose Production by the Purified Amy117

The purified Amy117 was incubated with pullulan solution in 50 mM sodium phosphate buffer (pH 7.0). The purity of the collected fractions was examined by TLC and HPLC, respectively.

The substrate concentration gradient experiment was conducted as follows: the dosage of substrate pullulan was 10 mg/mL, 40 mg/mL, 80 mg/mL, and 100 mg/mL, respectively, and the reaction was performed in a 2 mL system at 30 °C for 120 min, then the hydrolytic products were quantitatively analyzed by HPLC–ELSD (see Section 4.5). For the standard curves, 1, 2, 3, 5, 10, 25, 50, 100 mg/mL panoses were the abscissa values, and the corresponding characteristic peak areas were the ordinate values (Appendix A). The concentration of panose produced by the Amy117 was calculated according to the standard curve formula. The yield referred to the percentage of released panose weight (mg) to the initial total pullulan weight (mg).

### 4.7. Structural Simulation and Analysis

The structure simulation of Amy117 was first carried out by Alphafold2 [44], based on the sequence. Since two homologues, ThMA and BsNp (PDB: 1GVI and PDB: 1J0H), were determined as dimers [22,26], thus, YASARA software [45] was used for the homologous modelling with the Alphafold2-simulated Amy117 and the two homologues as templates. After the structures were obtained, the details of these structures were visualized by the Discovery Studio 2020 Client [46].

The sedimentation velocity experiments were performed in a ProteomeLab XL-I analytical ultracentrifuge (Beckman Coulter, Brea, CA, USA), equipped with AN-60Ti rotor (four-holes) and conventional double-sector aluminum centerpieces of a 12 mm optical path length, loaded with 380 μL of the protein samples and 400 μL of the buffer (50 mM Na_2_HPO_4_-NaH_2_PO_4_, pH 7.0). Before the run, the rotor was equilibrated for approximately 1 h at 20 °C in the centrifuge. Then, the experiments were carried out at 20 °C and 38,000 rpm, using the continuous scan mode and radial spacing of 0.003 cm. The scans were collected at 3 min intervals at 280 nm. The molecular weight and molecular weight distribution analysis of the polymers were performed using SEDFIT software (https://sedfitsedphat.nibib.nih.gov/software/default.aspx, accessed on 5 May 2022) and the continuous sedimentation coefficient distribution c(s) model [47]. The error of the molecular weight calculation was 5–10%.

The molecular dynamic simulations were performed using the AMBER14 force field, using YASARA software [45]. Firstly, the protein structure was imported into the YASARA for pretreatment, including hydrogenation, setting the environmental pH to the optimum pH of the protein, taking the protein as the center, setting the outward expansion distance to 5.0 Å as the dynamic simulation range, setting the simulation box to be periodic, and optimization of the energy. Then, the ion concentration, temperature, pressure, water density, force field, speed, and duration parameters were set to 0.9% NaCl (physiological solution), 303 K, 1 bar, 0.997 g/mL, AMBER14, normal speed (2 × 1.25 fs timestep), and 40 nanoseconds (ns), respectively. Before the simulation, the system automatically performs an equilibration to stabilize the temperature, pressure, and density. Finally, the molecular dynamics program was started and it automatically outputs the trajectory file of each frame (every 100 ps), which can be converted into the 3D structures diagrams for RMSD analysis [27,28].

## 5. Conclusions

In conclusion, a new *B. pseudofirmus* 703-origin cold-active neopullulanase Amy117 was discovered, capable of effective conversion of pullulan to panose. The optimum temperature of Amy117 is 30 °C, its activity is over 40% at 10 °C and over 80% at 20 °C, which helps to save energy and costs in industrial production. Amy117 can produce 72.7 mg/mL panose with a conversion yield of 91% (*w*/*w*), based on 80 mg/mL pullulan. Such high hydrolytic efficiency can greatly reduce the production costs of panose and avoid the costs of product separation, making it better suited to serve humans as an anti-caries sweetener, an anti-fading agent, antioxidant, and prebiotic.

## Figures and Tables

**Figure 1 ijms-23-06928-f001:**
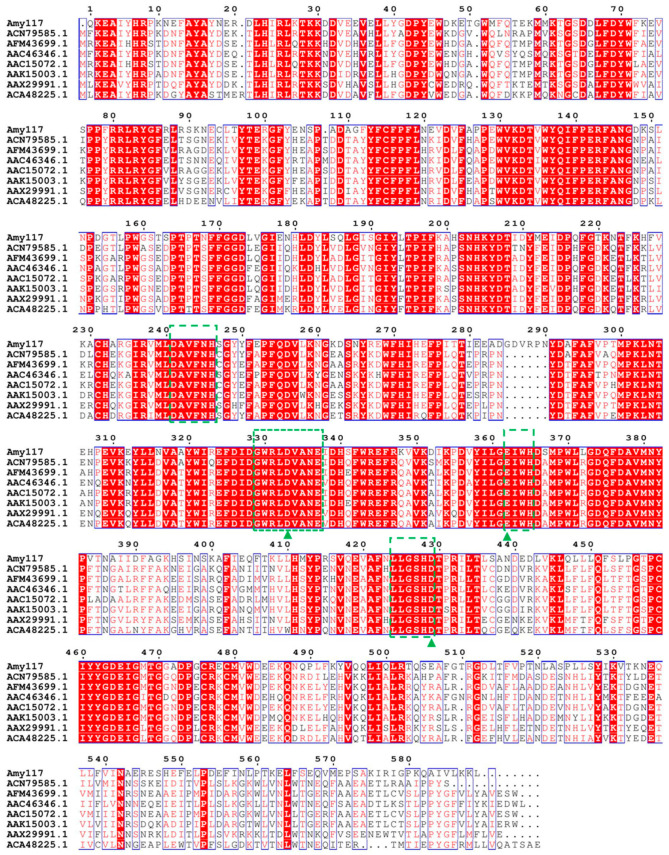
Sequence alignment of Amy17 from *B. pseudofirmus* 703 with some other bacterial α-amylases. The catalytic triads (DED) are marked with triangles, and four sequences conserved among enzymes belonging to the GH13 family are marked by green boxes.

**Figure 2 ijms-23-06928-f002:**
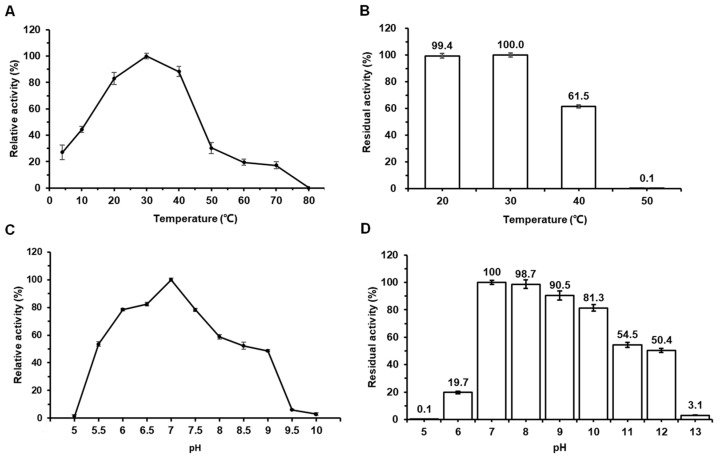
Effects of temperature and pH on the activity and stability of Amy117. (**A**) Effects of temperature on Amy117 activity. To determine the optimal temperature of Amy117, the reaction was conducted from 4 to 80 °C in sodium phosphate buffer (pH 7.0); (**B**) Effects of temperature on Amy117 stability. The thermal stability of Amy117 was determined at 20–60 °C in sodium phosphate buffer (pH 7.0) for 30 min. After incubation, the residual activity of Amy117 was measured at pH 7.0 and 30 °C; (**C**) Effects of pH on Amy117 activity. The reactions were conducted at 30 °C in different pH buffers; (**D**) Effects of pH on Amy117 stability. The stability of Amy117 was determined at different pH from 5.0 to 13.0 at 4 °C for 16 h. After incubation, the residual activity was measured at pH 7.0 and 30 °C. Each value of the assay was the arithmetic mean of triplicate measurements.

**Figure 3 ijms-23-06928-f003:**
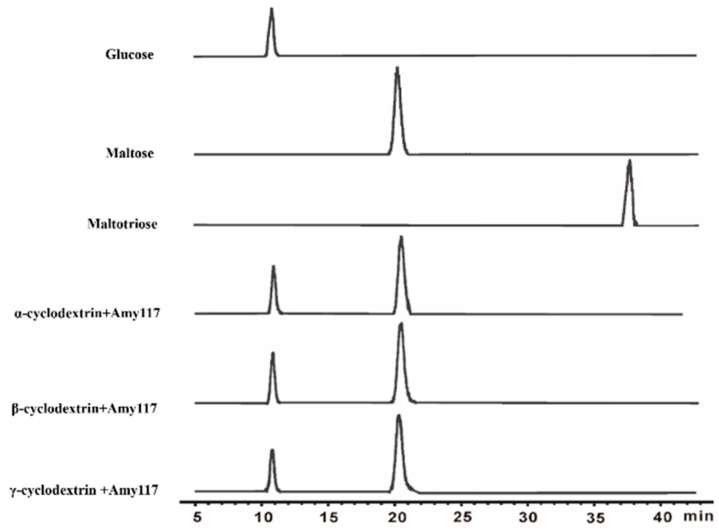
HPLC–ELSD chromatogram of hydrolysis products from α-cyclodextrin, β-cyclodextrin, and γ-cyclodextrin by Amy117. The separation was achieved using Agilent Zorbax Carbohydrate Analysis column (150 mm × 4.60 mm, 5 um). Mobile Phase: acetonitrile/water 75:25 (*v/v*), the flow rate was 1 mL/min. A drift tube temperature of 90 °C and a nebulizer gas flow rate of 2 L/min were used.

**Figure 4 ijms-23-06928-f004:**
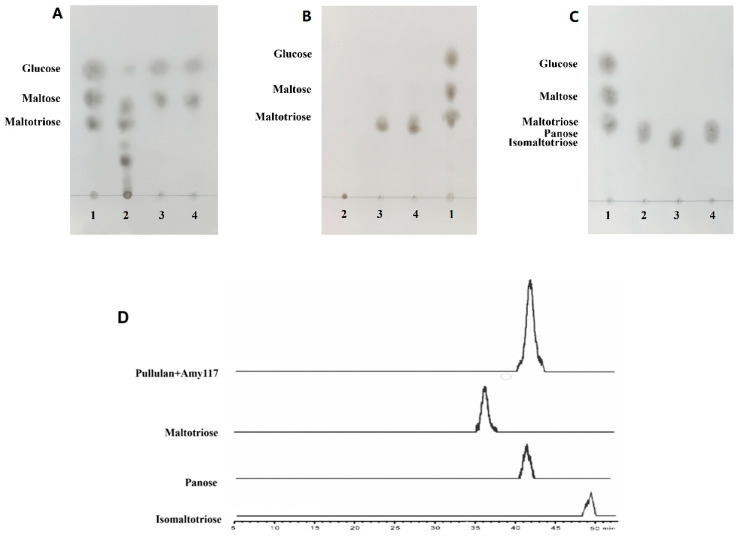
Analysis of hydrolysis products from soluble starch and pullulan. (**A**) Line1: standard mixture of glucose, maltose, and maltotriose; lines 2, 3, and 4: hydrolysate from soluble starch by BLA, Amy117, BLA, and Amy117 together; (**B**) Line 1: standard mixture of glucose, maltose, and maltotriose; lines 2, 3, and 4: hydrolysate from pullulan by BLA, Amy117, BLA, and Amy117 together; (**C**) Line 1: standard mixture of glucose, maltose, and maltotriose; lines 2 and 3: standard of panose, isomaltotriose; line 4: hydrolysate from pullulan by Amy117; (**D**) HPLC–ELSD chromatogram of hydrolysis products from pullulan by Amy117.

**Figure 5 ijms-23-06928-f005:**
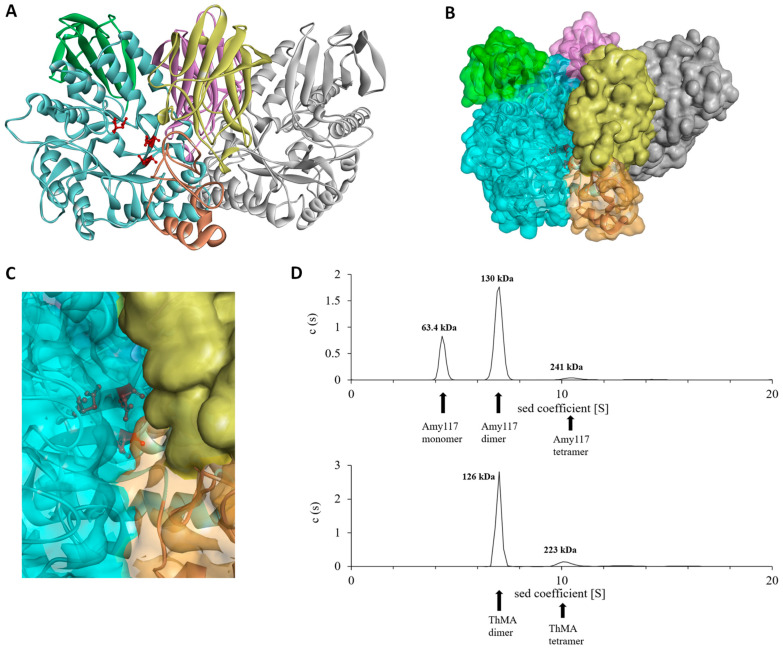
The simulated structure of Amy117. (**A**) The overall dimer structure of Amy117 is shown by the cartoon model; (**B**) Surface representation of Amy117; (**C**) Detailed view of the catalytic cleft. One monomer in the dimer structure is colored yellow (Domain N) and gray, and the domains N, A, B, C of the other monomer are colored purple, blue, brown, and green, respectively. Three catalytic residues, Asp333, Glu362, and Asp429, are presented as ball-and-stick models and colored red; (**D**) Analytical ultracentrifugation of Amy117 and ThMA.

**Figure 6 ijms-23-06928-f006:**
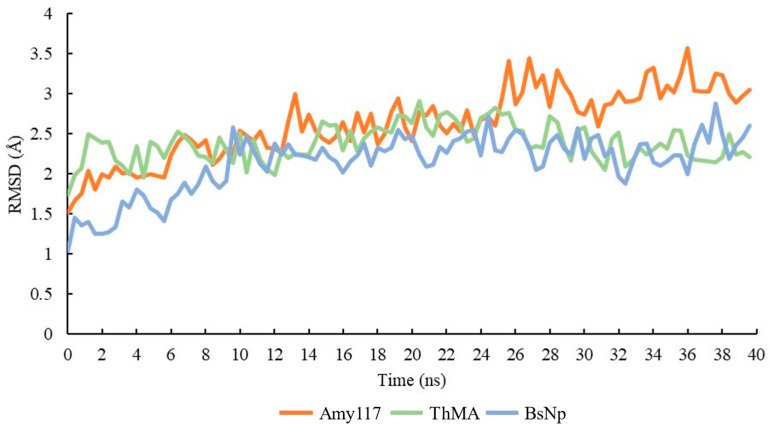
RMSD of Amy117, ThMA, and BsNp.

**Table 1 ijms-23-06928-t001:** Substrate specificity of Amy117.

Substrates ^1^	Specific Activity (U/mg)
α-cyclodextrin	851.92 ± 10.60
β-cyclodextrin	388.49 ± 1.33
γ-cyclodextrin	94.88 ± 0.34
Pullulan	62.50 ± 0.11
Soluble starch	29.31 ± 0.11
Amylopectin from maize	N.D. ^2^
Amylopectin from potato	N.D.
Amylose from potato	N.D.

All data are means ± SD (three biological replicates). LSD (least-significant difference) tests were used for multiple comparisons. Different letters above bars indicate that the means differ according to ANOVA and LSD tests (*p* < 0.01). Using glucose as the standard, the amount of enzyme that releases 1 μM of reducing sugar per minute under the measured conditions was called one unit (U) of α-glycosidase. The U/mg means enzymatic activity per milligram of protein, also known as specific enzymatic activity. ^1^ For the determination of substrate specificity, purified Amy117 was incubated in sodium phosphate buffer (50 mM, pH 7.0) at 30 °C with different substrates (1%, *w/v*); ^2^ N.D., no activity detected.

**Table 2 ijms-23-06928-t002:** HPLC–ELSD analysis of hydrolysis products from the different concentration of pullulan by Amy117.

Substrates (Pullulan)	Production of Panose (mg/mL)	Conversion Yield (*w*/*w*)
10 mg/mL	10.42	104.2%
40 mg/mL	40.33	100.8%
80 mg/mL	72.74	90.9%
100 mg/mL	72.72	72.7%

**Table 3 ijms-23-06928-t003:** The amino acid content of Amy117 and the other two proteins (ThMA and BsNp).

Number of Amino Acids	Arg	Arg/(Arg+Lys)	Val, Ile and Thr	Asn and Gln
Amy117	4.24%	0.40	4.58%	8.15%
ThMA	5.95%	0.53	3.23%	6.12%
BsNp	5.95%	0.55	4.08%	6.97%

## Data Availability

Not applicable.

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
