# Peer review of "Discovery of a New Microbial Origin Cold-Active Neopullulanase Capable for Effective Conversion of Pullulan to Panose"

_ijms, 2022, doi:10.3390/ijms23136928_

Round 1

Reviewer 1 Report

Dear Authors, the manuscript entitled ’ A cold-active and neutral neopullulanase capable of hydrolyzing high concentration pullulan to produce panose” is very interesting and has a high novelty impact. Nowadays a discovery a new biocatalyst that could be used for functional food production is a very important issue. The manuscript need an attention in a few places and some doubts need clarification. The suggestions that could improve the quality of the manuscript are listed below.

Comments;

1.       The title of manuscript could be slightly modified by adding words; discovery , characterization eg.

Characterization/Discovery a new microbial origin cold-active neopullulanase capable for effective conversion pullulan to panose

2.       Line 83-90. The clear aim of study without results and material and method should be presented.

3.       Line 93-117. The multiple sequences alignment or if possible structural alignment of homologs with an indication of conserved region position will be much better than the visualization of expression results by SDS-PAGE. The confirmation of purity (SDS-PAGE)q of recombinant protein should be rather moved to supplementary materials. I suggest using the Consurf server for visualization of the conserved regions on the recombinant protein model.

4.       On figure 2 c appears a point that suggests that activity was measured at 5.2 whereas other intervals are (0.5) different?. The formatting of plots style should be uniform for 2b and 2d

5.       Figure 4A-C the names of sugars should be written in big letters the same like at figure 4d.

6.       The reported substrate specificity assay of Amy117 is a large simplification. To be more accurate authors should determine a KM, Vmax and more significant Kcat

7.       Line 222-225; The reported result accuracy seem to be not significant and should be reduced to not more than two places after dot.

8.       Line 289-313. The author’s speculation about the correct name and classification of the analyzed enzyme seems to be out of the topic of our study. The highlighted part should be significantly reduced. The chronology of discussions does not correspond with the sequence of result presentations.

9.        Line 314-316; This part seems to repetition of discussion beginning

  Line 331-334; Authors tried explain the ability of Amy117 to be significantly active in low temperatures. However the explanation is focused on enzyme stability that not always fit well with optimal temperature for protein structure stability ?

11.   Line 340; The comparison of the Amy117 to ThMA and 1J0H is unclear. A short description of mentioned proteins should be added and appropriate references. The 1J0H is a PDB code of Bacillus stearothermophilus neopullulanase and this name should be used instead of signatures of records from databases.

12.   Line 400; Authors should precise the title of subsection.

13.   Line 401-402; Authors report that cannot provide definitive concentration of recombinant protein in solution. The best way for more specific assay of protein concentration of recombinant protein is measuring the solution  absorbance at 280 nm and calculate the it concentration with using molar extinction coefficient and molar mass. The value of molar extinction coefficient can be easily determined with using ProtParam tool on ExPaSy service. Authors also mention about protein concentration that is a result and should be avoided in materials and methods section.

14.   Line 356; instead of biochemical characteristic in case of enzyme with potential to be used in industry better use words; operational parameters and catalytic properties.

15.   The conclusion section could be more general with of course highlighting the main findings but with higher focus for possible applications in industry.

16.   Manuscript need intervention of English native speaker.

Reviewer 2 Report

In this manuscript a new neopullulanase was characterized and is considered a cold-active enzyme for panose production. 

-In the introduction a figure of the catalytic reaction is needed, as well as the chemical structure of panose and pullulan. In addition, data regarding the structure of the enzyme need to be added.

-Authors should explain the difference of enzyme specific activity for the different types of cyclodextrin. 

-Considering analytical ultracentrifuge, have been made any activity measurements to the fractions of protein with the different molecular weight?

-The spray that was used in TLC had only sulfuric acid and ethanol?

-In Figure 4, the legend of (D) is missing.

-In line 222, authors should add the specific activity of the enzyme that was used instead of concentration (mg/mL).

-In lines 141-142 authors mention that they incubated the enzyme in different pH values for 16 h, however in material and methods this time interval is 30 min (line 419).

Round 2

Reviewer 1 Report

Dear authors, the manuscript was significantly improved. After a small correction will be suitable to be published in IJMS.

 Comments;

 1.      Line 412-448. Please add the subsection for each analyzed parameter.

2.      Line 444; The sentence should rather sound like this;

The catalytic constants of Amy117 such as Km, Vmax and kcat  were determined by using 1.54 μM enzyme and varying concentrations of pullulan (0.5 mg/mL- 5.0 mg/mL) and soluble starch (1.0 mg/mL-10.0 mg/mL). All the measurements were done in triplicate. Finally the Km, Vmax and kcat were established by nonlinear regression using the GraphPad Prism software

3.      Line 449; Please change the subsection title; Analysis of composition the cyclodextrins enzymatic hydrolysis products by HPLC and TLC.